# Unravelling the link between SARS-CoV-2 mutation frequencies, patient comorbidities, and structural dynamics

Amirah Azzeri[1☯], Nurul Azmawati Mohamed[1], Saarah Huurieyah Wan Rosli[1], Muttaqillah Najihan Abdul Samat[2], Zetti Zainol Rashid[2], Muhamad Arif Mohamad Jamali[3], Muhammad Zarul Hanifah Md Zoqratt[4,5], Muhammad Azamuddeen Mohammad Nasir[4,5], Harpreet Kaur Ranjit Singh[4,5], Liyana Azmi[1☯]*

1 Faculty of Medicine and Health Sciences, Universiti Sains Islam Malaysia, Nilai, Negeri Sembilan, Malaysia, 2 Department of Medical Microbiology and Immunology, Faculty of Medicine, Universiti Kebangsaan Malaysia, Cheras, Kuala Lumpur, Malaysia, 3 Faculty of Science and Technology, Universiti Sains Islam Malaysia, Nilai, Negeri Sembilan, Malaysia, 4 Fast Genomics Solutions, Subang Jaya, Selangor Darul Ehsan, Malaysia, 5 School of Science, Monash University Malaysia, Bandar Sunway, Selangor, Malaysia

☯ These authors contributed equally to this work.
* liyana.azmi@usim.edu.my

**Data Availability Statement:** All relevant data are within the manuscript and its Supporting Information files.

## Abstract

Genomic surveillance is crucial for tracking emergence and spread of novel variants of pathogens, such as SARS-CoV-2, to inform public health interventions and to enforce control measures. However, in some settings especially in low- and middle- income counties, where sequencing platforms are limited, only certain patients get to be selected for sequencing surveillance. Here, we show that patients with multiple comorbidities potentially harbour SARS-CoV-2 with higher mutation rates and thus deserve more attention for genomic surveillance. The relationship between the patient comorbidities, and type of amino acid mutations was assessed. Correlation analysis showed that there was a significant tendency for mutations to occur within the ORF1a region for patients with higher number of comorbidities. Frequency analysis of the amino acid substitution within ORF1a showed that nsp3 P822L of the PLpro protease was one of the highest occurring mutations. Using molecular dynamics, we simulated that the P822L mutation in PLpro represents a system with lower Root Mean Square Deviation (RMSD) fluctuations, and consistent Radius of gyration (Rg), Solvent Accessible Surface Area (SASA) values—indicate a much stabler protein than the wildtype. The outcome of this study will help determine the relationship between the clinical status of a patient and the mutations of the infecting SARS-CoV-2 virus.

## Introduction

The coronavirus disease-2019 (COVID-19) was first isolated from Wuhan City, Hubei Province, China in December 2019. It is caused by Severe Acute Respiratory Syndrome Coronavirus 2 (SARS-CoV-2,) a novel betacoronavirus (β-CoV). The SARS-CoV-2 genome showed 79% homology to SARS-CoV, therefore possesses similarities of its pathogenesis,

**Funding:** "This study was funded by two the Universiti Sains Islam Malaysia Internal Research Grants: AA - Grant number PPPI/FPSK/0122/USIM/14622 and LA - Grant number PPPI/FPSK/0122/USIM/14322). The funders had no role in study design, data collection and analysis, decision to publish, or preparation of the manuscript."

**Competing interests:** The authors have declared that no competing interests exist.

epidemiology, viral origin, and mechanism of action to SARS-CoV [1]. Symptoms of SARS-CoV-2 infection vary widely in humans, ranging from mild flu-like symptoms such as fever, fatigue, and dry cough to complete respiratory failure [2]. Similar to severe acute respiratory syndrome (SARS), the transmission of COVID-19 is airborne, fomites, and air droplets [3]. The highly contagious nature of the virus prompted widespread lockdowns on a global scale to curb its spread.

In Malaysia, the first wave of COVID-19 infection with a total of 22 cases was first recorded on 25th January, 2020. Up until August 2023, there were a total of 5,121,858 COVID-19 cases with 37,165 deaths (COVID-19 | KKMNOW, 2023). The incidence of COVID-19 was highest amongst people in the age group of 55–64 and 63% of those >60 years were reported to be fatal. In addition, 81% of those above 60 years old had chronic comorbidities such as diabetes, and heart disease [4]. Recent studies have shown that immunocompromised patients with chronic diseases are more likely to harbour SARS-CoV-2 viruses with enhanced mutation rates [5, 6].

Coronaviruses, including SARS-CoV-2, exhibit significant genetic diversity and mutate rapidly. Amino acid substitutions within the viral structural proteins including S (spike), E (envelope), M (membrane), N (nucleocapsid), accessory proteins, and ORF1a/ORF1ab polyproteins can alter traits like pathogenicity and transmissibility, which also produce more virulent and infectious variants of SARS-CoV-2 [7, 8]. The emergence of more virulent and transmissible variants of SARS-CoV-2 has caused a surge in the number of cases and death tolls, collapsing many health systems. To address these challenges, frequent genomic surveillance is essential to track the emergence of new variants and their impacts. Shared virus sequences had been published in the Global Initiative on Sharing All Influenza Data (GISAID) database [9] to enable real-time genomic surveillance on a global scale [10]. As of August 2023, approximately 15 million full and partial genomes of SARS-CoV-2 have been submitted to GISAID. However, the number recorded in Malaysia is notably lower compared to other countries, potentially hindering the ability to detect and respond to emerging variants.

In relation to that, this study describes the associations between patient comorbidities with SARS-CoV-2 mutation types and numbers. We also analysed the mutation rates for SARS-CoV-2, particularly those residing within the ORF1a region. Finally, we investigated selected mutations of high frequencies and demonstrated via molecular dynamics, how the mutation potentially contributes to increased transmissibility of COVID-19.

## Methodology

### Sampling and ethics approval

Data analysis was conducted on retrospective 99 clinical histories of patients infected with COVID-19 from Hospital Canselor Tuanku Muhriz Universiti Kebangsaan Malaysia dating from 2nd June 2021 to 28th December 2021. The corresponding SARS-CoV-2 sequences of the infected patients were extracted from GISAID and the retrospective clinical histories of the patients matching to the SARS-CoV-2 sequences were collected. Each SARS-COV-2 sequence was matched to the patient's clinical history, which indicated the type of comorbidities diagnosed at the time of sampling for COVID-19. All the patient's information was made fully anonymous prior to any analysis. Ethics approval for the study was granted by the Research Ethics Committee, Universiti Kebangsaan Malaysia (JEP-2022-805).

### Mutation analysis

Global AY.59 genomes and metadata were obtained from the GISAID database [11] within the period of 2nd June 2021 until 28th December 2021. This is accessible with EPI_SET_230816ue

(10.55876/gis8.230816ue). Visualisation of the data analysis was done using Seaborn and Matplotlib Python packages [12, 13]. The phylogenetic tree of the global AY.59 was constructed based on the Nextstrain SARS-CoV-2 workflow using software IQ-TREE and Augur version 22.3.0 [14–16]. The phylogenetic tree was time-scaled using TreeTime [17]. Phylogenetic tree was visualised using baltic Python package (https://github.com/blab/baltic).

## Statistical analysis

Statistical analysis was performed using SPSS version 24.0 (SPSS Inc., Chicago, Illinois, USA). The results of descriptive analysis of sociodemographic characteristics, clinical characteristics, and the outcomes were presented for all patients and by disease stages. Visual assessment and normality tests, such as the Kolmogorov-Smirnov test, were used to test the normality of distribution prior to conducting and reporting the results of the descriptive analysis. Where appropriate, continuous variables were presented as means and standard deviations (SD). For findings that were not normally distributed, median, and inter-quartile range (IQR) was used. For categorical variables, results were presented as frequencies and percentages. The association between quantitative independent variable such as the number of comorbidities, and dependent variables were determined using Pearson's (Spearman's for non-normally distributed data) correlation test. Results for the bivariate analysis were presented with p-value and correlation coefficient when appropriate.

## Molecular dynamics

The nsp3 protease 3D structure was extracted from the protein databank (PDB) with the PDB ID: 7D7K. The mutation for the residue P822L (in the structure, was identified as residue 77) was performed using the mutagenesis function and visualised using PyMOL (The PyMOL Molecular Graphics System, Version 2.0 Schrödinger, LLC). The construct of P822L and wild type (WT) nsp3 system consisted of the protein in a solvated dodecahedron box with a minimum distance of 1.2 nm from the boundary. The systems were filled with single-point charge water; subsequently, it was neutralised by adding counter cations ($Na^+$) or anions ($Cl^-$) [18]. The solvated systems were then energy minimised for 5000 steps using the steepest descent method [19], followed by the equilibrium for 250 ps through number of particles (N), system volume (V), pressure (P) and temperature (T) ensembles to optimise the orientation and system density. The final equilibrated systems were used as starting conformation to run the MD simulations for 1000 ns. Finally, the output trajectories were obtained, and the estimation of Root Mean Square Deviation (RMSD), Root Mean Square Fluctuation (RMSF), Solvent Accessible Surface Area (SASA), and Radius of Gyration (Rg) was done using GROMACS packages. The graphs were analysed using XMGRACE software.

## Results

### Demography and correlation analysis

In this study, a total of 99 sequences were extracted from GISAID and matched to the patient clinical data dating from 2[nd] June 2021 to 28[th] December 2021. Among these sequences, 67 patients (67%) were females and 32 patients (32%) were males, with the majority falling within the age group of 30–39 years old (Fig 1A). Sorting the 99 SARS-CoV-2 genomes (S1 Table) by pangolin lineage revealed a notable dominance of AY.59 (61%), followed by AY.51 (13%), B.1.617.2 (11%), AY.79 (9%), and AY.42, B.1.351, AY.5, AY.114, and AY.53, each representing 1% (Fig 1B). We constructed a phylogenetic tree to then observe if there is a dominance of the AY.59 lineage in other countries (Fig 1C). Based on our phylogenetics, the AY.59 was shown

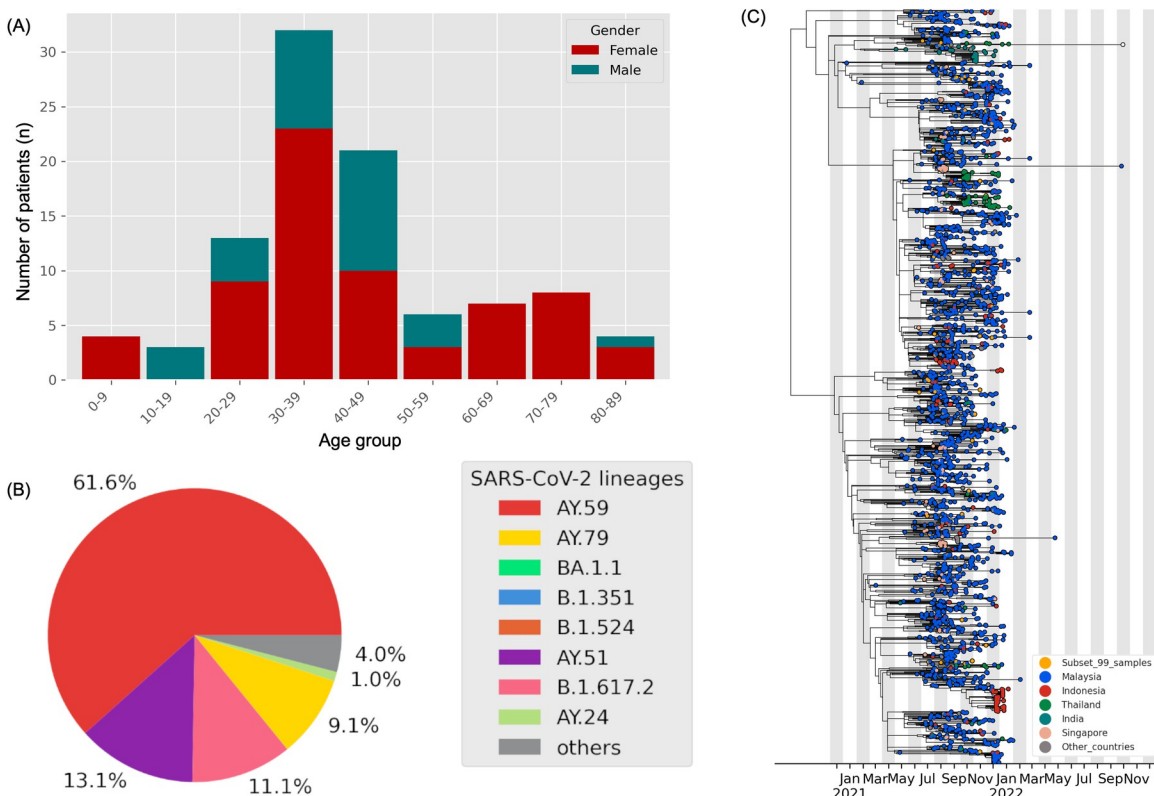

**Fig 1. Patient demography and phylogenetic distribution of SARS-CoV-2 from 2nd June 2021 to 28th December 2021.** (A) Age and sex distribution of our samples showed that the largest age group is 30–39 years old, with most samples being females (n = 67, 67%). (B) 61.6% of the circulating Pango lineage is represented by AY.59. (C) The phylogenetic tree represents the distribution of AY.59 in Malaysia, Indonesia, Thailand, Singapore, and other countries.

to dominate Malaysia, circulate mostly within the Southeast Asian regions and was only minimally detected in other parts of the globe.

## SARS-CoV-2 mutation analysis correlating with patient comorbidities

The patients included in this study were categorised based on the number of comorbidities diagnosed and the severity of their COVID-19 infections at the time of sample collection for SARS-CoV-2 sequencing. The type of comorbidities noted are those which are known to cause a reduction of immune status of a patient and to include the following: chronic heart, kidney, liver and lung diseases, diabetes melitus, dyslipidemia, and as well as inflammatory and infectious diseases, including HIV/AIDS. The number of comorbidities was classified according to clinical diagnoses, and the severity categories were recorded during the sampling period (Table 1). Among the categorised patients, the highest proportions were those with no known medical illnesses (46%), followed by patients with three or more comorbidities (27%), two comorbidities (10%), and a single comorbidity (16%). The severities of the patients were sorted based on COVID-19 categories with the following category 1 (asymptomatic) & 2 (symptomatic with no pneumonia), category 3 (symptomatic with pneumonia), 4 (symptomatic with pneumonia requiring supplemental oxygen) and 5 (critically ill with or without other organ failures) [20].

To determine the potential associations between SARS-CoV-2 mutations, patient comorbidities, and severities, correlation analysis was used accordingly. We investigated the number

**Table 1. Distribution of patient comorbidities and severities.**

| Groups | Total (n = 99) |
|---|---|
| **Comorbidity** | |
| 3 and above comorbidities | 27 (27%) |
| 2 comorbidities | 10 (10%) |
| 1 comorbidity | 16 (16%) |
| No known medical illness | 46 (46%) |
| **COVID-19 severity** | |
| Category 1 & 2 | 80 (80%) |
| Category 3, 4 & 5 | 19 (19%) |

of highest occurring mutations within the SARS-CoV-2 genome and mapped the frequency of mutations based on their respective genes/regions (Fig 2). The frequencies of amino acid substitutions revealed that the ORF1a gene had the highest number of amino acid substitutions (n = 92), followed by ORF1b (n = 51), S (n = 42), N (n = 20), M (n = 3), and finally E (n = 2) (Table 2).

Since ORF1a showed the highest number of occurring mutations, we performed a correlation analysis between the number of ORF1a mutations with the number of comorbidities. Our results indicated a significant link between the number of ORF1a mutations and the number of comorbidities (p value < 0.05). There were moderate-positive correlations between the number of comorbidities and number of ORF1a mutations, with higher number of comorbidities was associated with higher number of mutations [r = 0.05, p<0.05]. We also tested the correlation between COVID-19 severities with the number of ORF1a mutations. However, possibly, due to limited available data on the severity status of patients, our study found insignificant correlations between the number of mutations and the severity of COVID-19 infections.

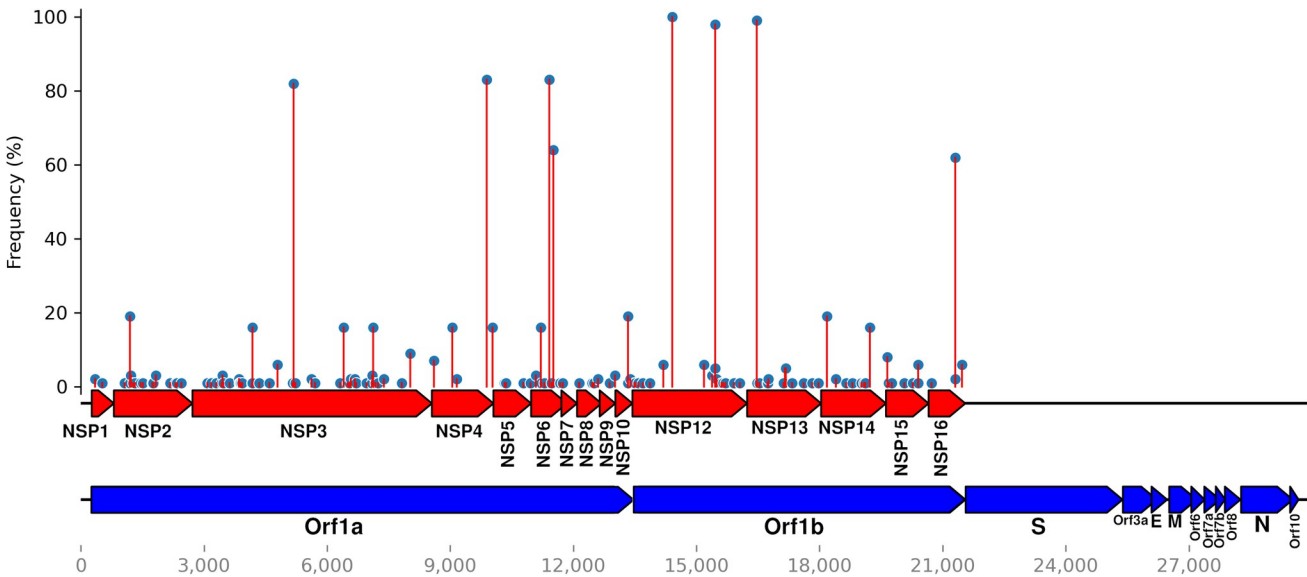

**Fig 2. Lollipop plot summarising the frequencies of SARS-CoV-2 amino acid substitutions in our study cohort (n = 99) in Table 2.** Red boxes represent genes and blue boxes represent coding sequences. A SARS-CoV-2 genome map with base-pair positions is displayed at the bottom. The bubbles in the Y-coordinates indicate mutation frequencies.

**Table 2. Genomic distribution of the identified SARS-CoV-2 mutations.**

| Gene/region | *n* of amino acid substitutions |
| --- | --- |
| ORF1a | 92 |
| ORF1b | 51 |
| S | 42 |
| N | 20 |
| M | 3 |
| E | 2 |

## ORF1a mutation profiles

The ORF1a gene encodes essential non-structural proteins that are crucial for the viral replication machinery and the maintenance of the viral genome. Adaptive mutations occurring in ORF1a/b might enhance viral replication, drug resistance, and increased virulence. We also questioned if mutations within this region could lead to the enhanced ability of SARS-CoV-2 to infect a patient with comorbidities. Since ORF1a displayed the highest number of mutations, we further analysed the mutation locations within the gene and sorted them based on their frequencies for each patient (Fig 3). Mutation analysis of the ORF1a region revealed up to 92 amino acid (AA) substitutions in our samples. Sorting of the mutation frequencies revealed the top 5 frequent mutations occurred to be nsp6 V149A, nsp4 A446V, nsp3 P822L, nsp6 T181I and nsp2 P129L (Table 3).

Since the mutation nsp3 P822L (hereon referred to as P822L) encodes for the papain-like protease (PLpro), we were interested to know if there was a correlation between this mutation and patient comorbidities, severity as well as infectivity. We were interested to know if P822L was correlated with patient comorbidities. Do patients harbouring SARS-CoV-2 with P822L cause enhanced COVID-19 severity and are more transmissible? To answer these questions, we performed a correlational analysis of P822L with patients with comorbidity status. Our results indicate trends that suggest higher comorbidities and severe COVID-19 for patients harbouring P822L. However, our results were insignificant due to the small sample size.

## Molecular dynamic simulation of P822L

Notably, the PLpro plays a crucial role in facilitating viral replication by cleaving the ORF1ab polyprotein into functional segments (nsp1-nsp3). Additionally, it serves as a mechanism for the virus to evade the host's immune response by eliminating the interferon-stimulating gene-15 protein (ISG15) from host proteins, thus disrupting the host's innate antiviral defence. Previous studies suggest that mutations in PLpro can lead to alterations in the enzyme's specificity and, as well as, cause reductions in antiviral effectiveness [21].

The potential impact of the P822L mutation on PLpro stability was tested using molecular dynamics (MD) simulations on the apo structure of the SARS-CoV-2 PLPro protease. However, seeing that the active form of PLpro is a homodimer relating to its biological activity, we chose to perform the simulations on the bat SARS-CoV PLpro homodimer, BtSCoV-Rf1.2004 (PDB ID: 7SKQ). Furthermore, PLPro from BtSCoV-Rf1.2004 shares high sequence similarity with PLpro from SARS-CoV-2 (82%) [22] and warrants our basis for selecting the homodimer form of PLpro for analysis. Using PyMOL, we generated a model containing the mutant P822L (Fig 4). To assess protein stability, MD simulations of the WT and P822L were performed over a 1000 ns timescale. Parameters including RMSD, RMSF, and Rg were employed for analysis.

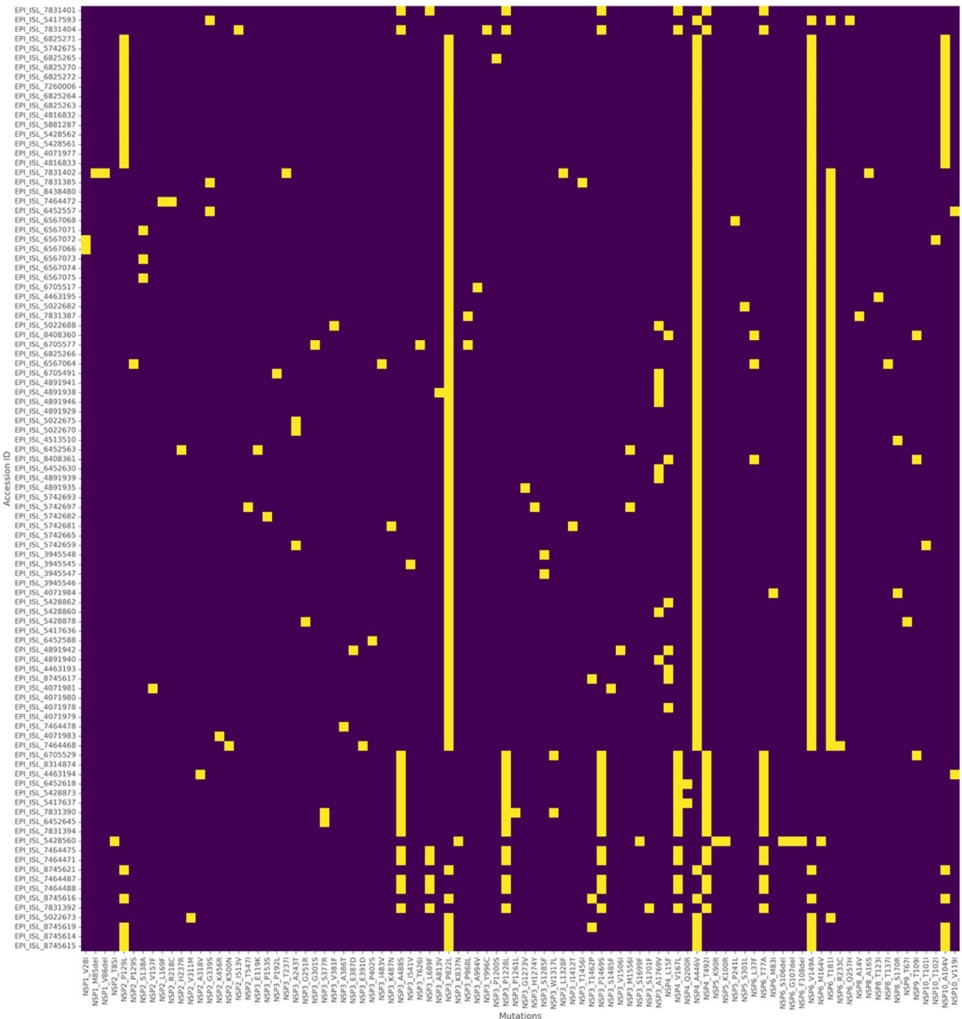

**Fig 3. Frequency of mutations in ORF1a.** The Y-axis lists the number of patient samples and X-axis lists the frequency and type of mutations per patient.

The RMSD analysis is a measure of protein stability. The RMSD plot (Fig 5A) indicates that P822L displayed significantly lower RMSD values compared with the WT. The WT, displayed high fluctuations, especially within the 0–120 ns timepoint, but then reached a latent phase, indicating stabilising conformations. Comparing both systems, the lower RMSD values for P822L suggests that the mutation might have reduced the protein's flexibility, potentially enhancing its resistance to environmental changes. The average RMSD values throughout the entire simulation times for P822L (1.68 nm) was lower compared to WT (2.86 nm).

**Table 3. The top five occurring mutations within ORF1a.**

| Gene | AA substitution | No. of occurrence (n) | Frequencies (%) |
| --- | --- | --- | --- |
| nsp6 | V149A | 82/99 | 82.8 |
| nsp4 | A446V | 82/99 | 82.8 |
| nsp3 | P822L | 81/99 | 81.8 |
| nsp6 | T181I | 63/99 | 63.6 |
| nsp2 | P129L | 19/99 | 19.1 |

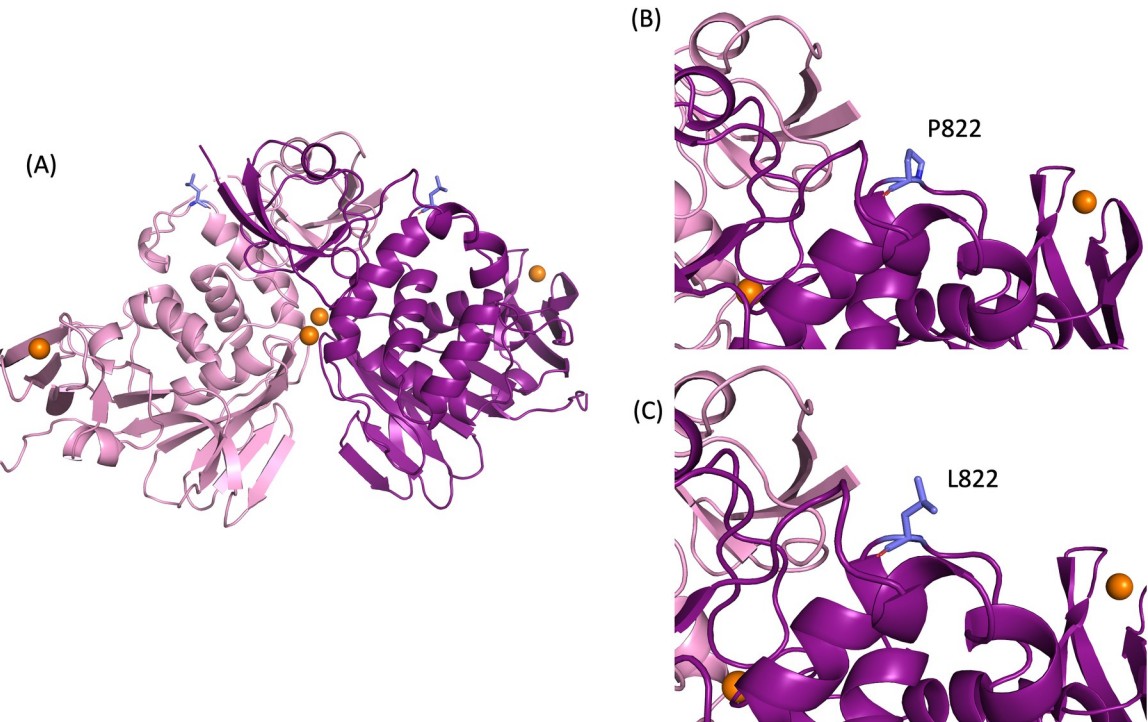

**Fig 4. Generation of P822L in PLpro.** (A) Apo form of Bat SARS-CoV PLpro homodimer represented in cartoon with zinc (orange spheres) The P822L was generated and visualised using PyMOL. Here, we present a (B) zoomed view of WT residue P822 and (C) mutated L822.

Additionally, since the WT possesses a higher maximum RMSD, the energy landscape and conformational changes within the WT is more significant compared with P822L.

Since the RMSD differs between both systems, we investigated the SASA to gain insights on their conformational changes and accessibility to solvents. Our SASA analysis showed comparable SASA fluctuations for both systems. The average SASA values for both WT and P822L are 311.82 $nm^2$ and 314.67 $nm^2$, respectively.

Based on the comparable SASA values, we also analysed Rg values to assess the compactness and rigidity of both proteins. The WT and P822L exhibited comparable fluctuations during the first 50 ns. However, the WT structure showed spikes of fluctuations at 100 ns,200 ns, 550 ns, and 700 ns before dipping on the 800 ns onwards. Compared with the WT structure, the mutated version displayed notably stable values throughout the simulation, suggesting a potentially more stable system (Fig 5C). However, the larger Rg values for P822L indicated a less compact structure and may affect the structural dynamics of the homodimer.

We then assessed the RMSF complex to understand the flexibility of both structures. RMSF involves observing the C-alpha atom of the models' residues to infer each atom's fluctuations across the C-alpha backbone. Based on the RMSF simulation (Fig 5B), the change of P822 to L822 showed fluctuation, as observed in residue 77. We performed RMSF on both chains of the WT and P822L. In short, within chain A, the RMSF profile remains comparable between both WT and P822L. Interestingly in chain B, P822L demonstrates higher fluctuations within two areas, (residues H51 and Y72).

## Discussion

Our genomic analysis of 99 samples revealed a prominent prevalence of the Pango AY.59 lineage during Malaysia's third wave of COVID-19. Interestingly, another study noted a

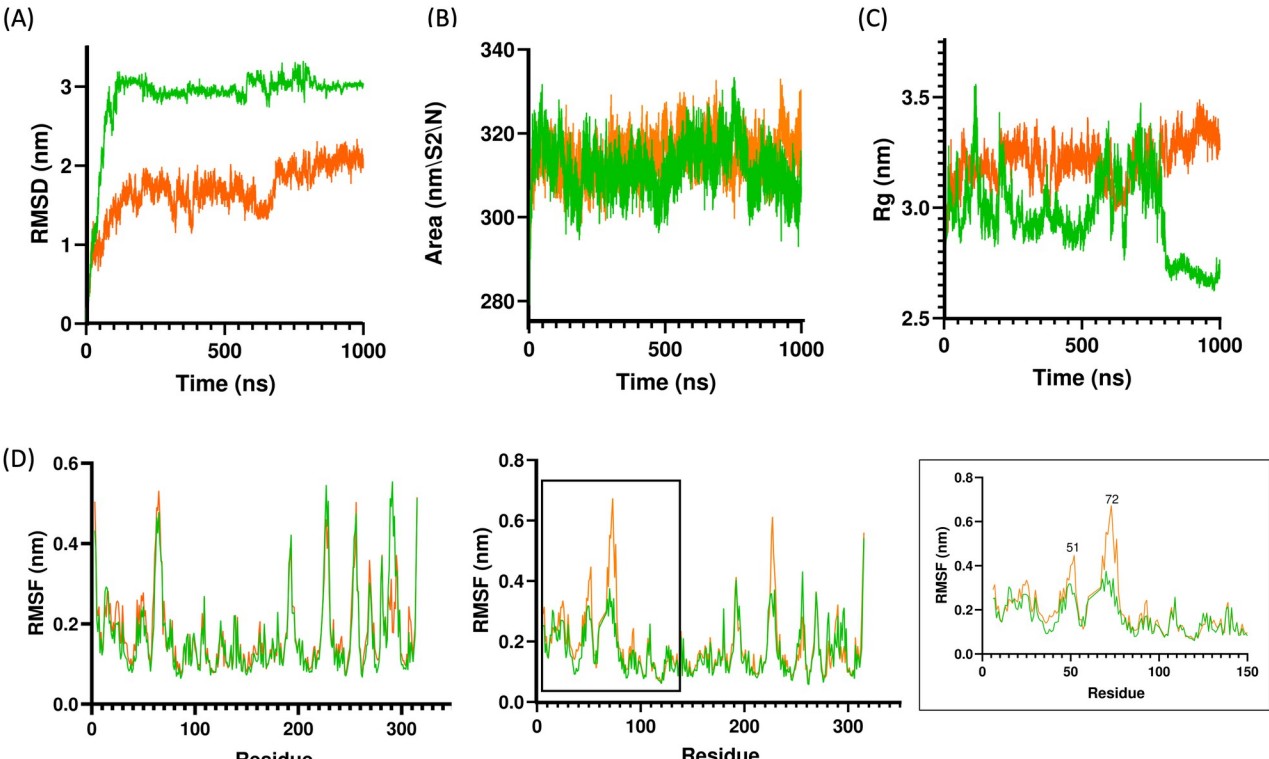

**Fig 5. Molecular dynamic simulations of WT and P822L.** The stability of the WT apo form of Bat SARS-CoV PLpro homodimer (green) and P822L (orange) were evaluated using (A) RMSD, (B) SASA, (C) Rg and (D) RMSF simulations, which was plotted for both chain A (left) and chain B (right). In Chain B, a zoomed view of residues 3–150 was provided to highlight the fluctuations of the P822L mutant. Our simulations demonstrate that P822L is stabler than the WT.

dominance of AY.79 in West Coast areas of Malaysia, indicating the infection patterns of the infected patients between June–December 2021 [22]. Notably, the Delta variant is notable for its heightened virulence and prompted us to explore the correlation between patient comorbidity levels and the manifestation of SARS-CoV-2 mutations.

Various factors contribute to COVID-19 transmission, including patient demographics that include sex, age, and ethnicity, which may impact disease outcomes. In this study, we investigated the number of patient comorbidities that could play a role in manifesting SARS-CoV-2 mutations. Are patients with more comorbidities more favourable hosts for viral replication, leading to beneficial mutations for viral transmissibility? Our study has revealed a significant link between higher comorbidity counts and enhanced mutations within the SARS-CoV-2 virus. Previous studies showed that comorbidities that are associated with higher ACE2 expression may enhance the virus entry and the severity of COVID-19 infection [23]. It is possible that comorbidities contribute to the enhanced systemic inflammation releases reactive oxygen species and drives mutations. Changes in the biochemical process have been shown to induce errors in replication, editing, or damage to a nucleic acid [24]. Regarding the sex ratio of our dataset, we utilised a convenience sampling approach, thus demonstrating an unbiased scenario of COVID-19 patients of Universiti Kebangsaan Malaysia within the specified time. However, our dataset featured a 2:1 female-to-male ratio. We performed a correlation analysis between gender and the frequency of amino acid substitutions. However, there was no statistically significant association between gender and the number of mutations, although males reported higher mutations than females [p = 0.769]. Based on this trend, it is

likely that males are predisposed towards generating higher frequencies of SARS-CoV-2 mutations. However, a bigger sample size is warranted to fully comprehend the significance of sex with mutation frequencies for SARS-CoV-2. Notably, the rate of viral mutations also depends on community characteristics like sex, age [25], ethnicities [26], and host genetic variabilities [27, 28]. Furthermore, the different types of selective pressures determine the types of mutations to occur. A study by Wilkinson et al., (2022) [29] showed that in long-term infections, there is a tendency to partly select for mutations which aid the virus with *intra*-host replication (cell-to-cell transmission) and persistence as opposed to the general SARS-CoV-2 population, where mutations, which aid *inter*-host transmission are more strongly selected. Work by Maurya et al. (2022) [30] supported our findings, as they identified a single mutation (S194L) to frequently occur in their mortality group. This implies the exclusivity or tendency for mutations to occur in patients with severe disease progressions can be observed.

Regarding COVID-19 severity, when we performed a correlational analysis to explore the potential link between COVID-19 severity and the number of patient comorbidities, we found the connection between these factors to be statistically insignificant. Notably, other studies have shown that patient immune status may have a role in diversifying mutations in SARS-CoV-2. For example, a case report from Hensley et al., (2021) [31] showed prolonged SARS-CoV-2 infection in a patient with multiple myeloma. Extraction and genomic profiling of the virus demonstrated high viral replication and diversification within the patient prior to the patient's death. Potentially, future studies could demonstrate the relationship between COVID-19 severities and SARS-CoV-2 mutations by adopting a larger sample size. We also noted that our study only considered the number of comorbidities and did not consider the type of comorbidities with the number and type of SARS-CoV-2 mutations. Particularly, hypertension, diabetes mellitus, and coronary artery diseases have been shown to contribute to disease severity and susceptibility for in patients to be infected by SARS-CoV-2 [32]. Future work can further delve into the type of comorbidities and associate this with the type of mutations within SARS-CoV-2.

The increase in undiagnosed COVID-19 mutations in people with comorbidities poses a serious public health risk. Comorbidities compromise the immune system and predisposes a person with severe illnesses when they contract the virus. When combined with highly transmissible variants such as Delta and Omicron, the impact can be particularly severe, leading to higher rates of infection in the population [33]. This could lead to increased demand on health systems, which could be overwhelmed by an increase in severe cases, putting pressure on medical resources and health professionals. In addition, persons with comorbidities who were infected with a variant of concern were more likely to require hospitalisation and intensive care [34]. This highlights the importance of vigilant surveillance and detection of variants, especially in populations with pre-existing health conditions.

The presence of undetected variations in people with comorbidities may affect the effectiveness of vaccination. Variants with mutations that allow them to partially bypass immunity, such as Omicron, may reduce the protective benefits of vaccination and previous infections. This increases the risk of severe disease in both vaccinated and unvaccinated patients with concomitant disease. A multi-pronged strategy is needed to reduce the burden of undetected variation in patients with comorbidities. This includes increased genomic surveillance to detect new variations and investigate their potential impact on disease severity and vaccine efficacy. In addition, targeted public health interventions [35], such as promoting vaccination campaigns to people with comorbidities, can help reduce the overall burden of the disease within this vulnerable group.

Following SARS-CoV-2 transmissibility, we investigated the second highest number of mutations within the samples, which is the P822L of the PLpro. The PLpro domain of nsp3 is a

highly conserved domain, which encodes for host cell survival signalling pathways [36] and thus implies lower mutation rates within this region compared to other viral regions such as the spike protein. However, it was interesting to note that for our tested samples, the PLpro showed high mutation frequencies of nsp3 P822L. Abbasian et al., (2023) [37] reported high mutation rates (over 50%) in the nsp3 region. On the other hand, Anwar et al., (2022) [38] described P822L to be a single occurring mutation of their tested samples. Interestingly, the recurrent mutations of P822L were also identified in immunodeficient patients [29], supporting the notion that a host's genetic and/ or disease status do play a role in fostering beneficial mutations for SARS-CoV-2.

We questioned if the P822L might play a role in attenuating or increasing viral fitness or infectivity. Throughout the virus's evolution, various mutations in the same residue occurred, one of them being ORF1a P1640S [37]. This suggests that these mutations on this site led to the stability of the protein folding and could confer enhanced virulence of SARS-CoV-2. Naderi et al., (2023) [39] showed that nsp3 protease localises to the deubiquitinating site in the PLpro domain which overlaps with the ISG15 binding site, suggesting it may modulate the host's antiviral responses. Indeed, in our study, we show that the P822L is a stabilising mutation, albeit some changes in its structural dynamics. Overall, our results indicate a protein with potentially improved function by suppressing the impact of other deleterious mutations [40]. We tracked the P822L mutation across lineages and found that P822L from the Delta lineage is retained and continues to occur in the Omicron lineage [41]. Furthermore, our RMSD data, indicating fluctuations post 550 ns, highlights the importance of extended simulations in other systems. Future *in vitro* experiments can provide valuable insights into the role of P822L towards nsp3 stability.

## Conclusion

Genetic surveillance is a crucial tool for scouring variants and analysing infection patterns within populations. Furthermore, the availability of an open-sourced database with genomic information enabled us to assess the role of patient status such as comorbidities in COVID-19 infections with SARS-CoV-2 mutations. Future studies should include larger sample sizes to assess the role of SARS-CoV-2 mutations with COVID-19 severity. Notably, the type of SARS-CoV-2 mutation also, could potentially affect the type and number of patient comorbidities and severity of the COVID-19 infections. We showed trends suggesting a possible association between P822L and higher comorbidity rates and severe COVID-19 outcomes, but it was limited due to the small sample size. Notably, the nsp3 P822L mutation of the PLpro domain exhibited one of the highest mutation frequencies, suggesting its potential role in viral replication enhancement and virulence.

Our molecular dynamics simulations indicated that P822L exhibited increased stability compared to the WT structure, potentially resulting in enhanced resistance to environmental fluctuations. The fluctuation analysis further revealed that P822L might have led to reduced flexibility in certain regions of the protease, potentially affecting its function. Overall, our findings underscore the need for more extensive research and larger datasets to elucidate the intricate connections between SARS-CoV-2 mutations, patient characteristics, and disease outcomes.

In summary, our study provides valuable insights into the genetic diversity of SARS-CoV-2 genomes, sheds light on potential correlations between mutations and comorbidities, and introduces intriguing implications of the nsp3 P822L mutation through molecular dynamics simulations. While our findings hold promising avenues for further investigation, their significance awaits confirmation through larger-scale studies and enhanced data availability.

## Supporting information

**S1 Table. Patient comorbidities including number of comorbidities, stage of COVID-19 infection and sequenced lineage of SARS-CoV-2 for each corresponding patient.** (PDF)

## Acknowledgments

We gratefully acknowledge all SARS-CoV-2 data contributors, i.e., the Authors, their originating and submitting laboratories responsible for obtaining the specimens, including Mohd Noor Mat Isa from Malaysia Genome and Vaccine Institute (MGVI), Universiti Kebangsaan Malaysia, and Nor Azila Muhammad Azami from Universiti Kebangsaan Malaysia Institute Molecular Biology for generating the genetic sequence, metadata and sharing via the GISAID Initiative, on which this research is based. We would also like to acknowledge Dr. Azima Abdul Aziz from Universiti Putra Malaysia for the proof reading of this manuscript. Finally, we would like to extend our appreciation to Associate Professor Dr. Amir Syahir Amir Hamzah from Universiti Putra Malaysia for his assistance in providing the server for our molecular dynamic simulations. This research was funded by USIM Internal Grant, grant number PPPI/FPSK/0122/USIM/14622 and PPPI/FPSK/0122/USIM/14322. Ethics approval by Research Ethics Committee, Universiti Kebangsaan Malaysia (JEP-2022-805).

## Author Contributions

**Conceptualization:** Amirah Azzeri, Liyana Azmi.

**Data curation:** Amirah Azzeri, Muhammad Azamuddeen Mohammad Nasir, Liyana Azmi.

**Formal analysis:** Amirah Azzeri, Muhamad Arif Mohamad Jamali, Muhammad Zarul Hanifah Md Zoqratt, Liyana Azmi.

**Funding acquisition:** Amirah Azzeri, Liyana Azmi.

**Investigation:** Amirah Azzeri, Nurul Azmawati Mohamed, Zetti Zainol Rashid, Muhammad Zarul Hanifah Md Zoqratt, Muhammad Azamuddeen Mohammad Nasir, Harpreet Kaur Ranjit Singh, Liyana Azmi.

**Methodology:** Amirah Azzeri, Muhamad Arif Mohamad Jamali, Muhammad Zarul Hanifah Md Zoqratt, Harpreet Kaur Ranjit Singh, Liyana Azmi.

**Project administration:** Nurul Azmawati Mohamed, Muttaqillah Najihan Abdul Samat, Zetti Zainol Rashid, Muhammad Azamuddeen Mohammad Nasir, Liyana Azmi.

**Resources:** Amirah Azzeri, Saarah Huurieyah Wan Rosli, Zetti Zainol Rashid, Muhammad Azamuddeen Mohammad Nasir, Harpreet Kaur Ranjit Singh, Liyana Azmi.

**Software:** Muhamad Arif Mohamad Jamali, Muhammad Zarul Hanifah Md Zoqratt, Liyana Azmi.

**Validation:** Amirah Azzeri, Nurul Azmawati Mohamed, Saarah Huurieyah Wan Rosli, Muttaqillah Najihan Abdul Samat, Zetti Zainol Rashid, Muhamad Arif Mohamad Jamali, Liyana Azmi.

**Visualization:** Muhamad Arif Mohamad Jamali, Muhammad Zarul Hanifah Md Zoqratt, Liyana Azmi.

**Writing – original draft:** Amirah Azzeri, Nurul Azmawati Mohamed, Saarah Huurieyah Wan Rosli, Zetti Zainol Rashid, Muhammad Zarul Hanifah Md Zoqratt, Harpreet Kaur Ranjit Singh, Liyana Azmi.

**Writing – review & editing:** Amirah Azzeri, Nurul Azmawati Mohamed, Muttaqillah Najihan Abdul Samat, Zetti Zainol Rashid, Muhamad Arif Mohamad Jamali, Muhammad Zarul Hanifah Md Zoqratt, Muhammad Azamuddeen Mohammad Nasir, Harpreet Kaur Ranjit Singh, Liyana Azmi.

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
