## [Decision Letter · Decision Letter 0]

29 Dec 2023

PONE-D-23-28307Unravelling the Link between SARS-CoV-2 Mutation Frequencies, Patient Comorbidities, and Structural DynamicsPLOS ONE

Dear Dr. Azmi,

Thank you for submitting your manuscript to PLOS ONE. After careful consideration, we feel that it has merit but does not fully meet PLOS ONE’s publication criteria as it currently stands. Therefore, we invite you to submit a revised version of the manuscript that addresses the points raised during the review process.

We look forward to receiving your revised manuscript.

Kind regards,

Ahmed A. Al-Karmalawy, Ph.D.

Academic Editor

PLOS ONE

Journal Requirements:

4. We note that there is identifying data in the Supporting Information file <Supporting information.pdf>. Due to the inclusion of these potentially identifying data, we have removed this file from your file inventory. Prior to sharing human research participant data, authors should consult with an ethics committee to ensure data are shared in accordance with participant consent and all applicable local laws.

-Location data

Please remove or anonymize all personal information (collection date), ensure that the data shared are in accordance with participant consent, and re-upload a fully anonymized data set. Please note that spreadsheet columns with personal information must be removed and not hidden as all hidden columns will appear in the published file.

Reviewers' comments:

Reviewer's Responses to Questions

**Comments to the Author**

1. Is the manuscript technically sound, and do the data support the conclusions?

Reviewer #1: Yes

Reviewer #2: Yes

2. Has the statistical analysis been performed appropriately and rigorously? 

Reviewer #1: Yes

Reviewer #2: Yes

3. Have the authors made all data underlying the findings in their manuscript fully available?

Reviewer #1: Yes

Reviewer #2: Yes

4. Is the manuscript presented in an intelligible fashion and written in standard English?

Reviewer #1: Yes

Reviewer #2: Yes

5. Review Comments to the Author

Reviewer #1: Authors of the presented manuscript introduced an epidemiological study associating patient comorbidities with SARS-CoV-2 mutation types and numbers, focusing their investigations on the most frequent ones at the ORF1a gene region. Furthermore, they highlighted the impact of particular mutation on the thermodynamic stability of PLpro enzyme. The manuscript is relevant, valuable in the field, and with potentiality for high citation. Here are some points should be considered prior publication.

1. The authors adopted the PDB.file (7D7K) for performing the computational analysis which is deposited in its monomeric state. Typically, the PLPro is a homodimer relating to its biological activity (https://doi.org/10.1021/acsinfecdis.1c00631) and so computational study is better within its homodimeric state since it is the active form of the protein. Authors should provide a rational for adopting monomer state for computational analysis.

2. Authors should provide a rational for performing computational analysis on an adopted apo over holo state of the PLpro protein.

3. Authors are advised to better schematically represent Table 2 highlighting the location of the gene regions.

4. Authors announced their monitoring for the molecular dynamics SASA trajectories; however, this figure is missing. It would be interesting to visualize such data.

5. In Figure 3A, it is better to highlight the mutated residue first, prior zooming in.

Reviewer #2: Good job doing this research in a hot and important topic.

The authors carried out a meta-analysis of the SARS-CoV-2 patients' data and their respective viral genomic diversity. Their work showed that patients comorbidities are significantly linked with more liability of the infecting SARS-CoV-2 virus to mutate towards more viral virulence, infection persistence and potential enhanced transmissibility. This work would hopefully be a primer for other larger studies with more significant implications.

My comments are summarized in these points, though I attached a copy of the manuscript with highlights and comments to point at them:

1- All dates need to be revised thoroughly, either those of this study at their different occurrences in the manuscript and of other cited work.

2- The manuscript is well written in terms of the language and clarity of statements. However, some words need to be revised and modified for further clarity, notably:

a. Line 34 -> “persistent”.

b. Line 151 -> “Genomic distribution of amino acid substitutions ...”.

c. Line 168 -> “contains”.

d. Line 257 -> “occur”.

e. Line 353 -> “demography”.

3- In line 103, There are unresolved abbreviations that need to be elucidated. The abstract should have the full wording of any abbreviations as it is mostly read separately.

4- In line 139 in the Results section, the percentages summarized in text from Table 1 were different than those in the table, so please revise.

5- In Figure 1 and more than once in the text, Age and sex were included as important factors in the rate of viral mutation (as in line 114 in the Results and line 259 in the Discussion). However, the significance of these factors was not addressed in the study which was carried out using a dataset with a 2:1 female to male ratio. The authors need to show how sex could impact their results.

6- In line 265 in the Discussion section, the statement “work of ...... supported our findings”, is somehow confusing as the reader here may need to know which findings in particular are consistent with the cited reference.

Please, refer to the attached manuscript pdf for further clarification of the points.

6. PLOS authors have the option to publish the peer review history of their article (what does this mean?). If published, this will include your full peer review and any attached files.

Reviewer #1: **Yes**

Reviewer #2: **Yes**

---

## [Author Response · Author response to Decision Letter 0]

19 Jan 2024

Reviewer 1

Comment: The authors adopted the PDB.file (7D7K) for performing the computational analysis which is deposited in its monomeric state. Typically, the PLPro is a homodimer relating to its biological activity (https://doi.org/10.1021/acsinfecdis.1c00631) and so computational study is better within its homodimeric state since it is the active form of the protein. Authors should provide a rational for adopting monomer state for computational analysis.

Response: It was an oversight from our side that we performed our computational analysis of the PLPro in monomer state. To that end, we would like to replace the monomer results with the homodimer form.

In the original manuscript, we show that the mutated form is more stable compared with the WT. When we performed the homodimer analysis, this state was much more amplified and adds to the significance of our results. To this end, we have presented our new results and discussions in the updated manuscript.

Additionally, we have added a rational for choosing the homodimer form of PLPro in the following sentence:

“However, seeing that the active form of PLpro is a homodimer relating to its biological activity, we chose to perform the simulations on the bat SARS-CoV PLpro homodimer, BtSCoV-Rf1.2004 (PDB ID: 7SKQ). Furthermore, PLPro from BtSCoV-Rf1.2004 shares high sequence similarity with PLpro from SARS-CoV-2 (82%) (22) and warrants our basis for selecting the homodimer form of PLpro for analysis”. We would like to once again, thank the reviewer for this insightful comment.

Comment: Authors should provide a rational for performing computational analysis on an adopted apo over holo state of the PLpro protein.

Response: We chose to adopt the apo state of PLPro over its holo state to elaborate its conformational changes prior to binding to its ligands. Our results show that the WT form exhibits high RMSD and reflects conformational changes, which might not be visible should we opt for its holo form.

Comment: Authors are advised to better schematically represent Table 2 highlighting the location of the gene regions.

Response: We agree with the reviewer that Table 2 alone would not represent our data best. To this end, we have added a lollipop plot (Fig 2) to support Table 2, which also maps out the frequencies and location of the amino acid substitutions. The figure was highlighted in this sentence:

“We investigated the number of highest occurring mutations within the SARS-CoV-2 genome and mapped the frequency of mutations based on their respective genes/regions (Fig 2).”

Comment: Authors announced their monitoring for the molecular dynamics SASA trajectories; however, this figure is missing. It would be interesting to visualize such data.

Response: We apologise for the oversight on our part, and have included and discussed the significance of the SASA trajectories for both systems. We express our gratitude to the reviewers for their thorough analysis of this section.

Comment: In Figure 3A, it is better to highlight the mutated residue first, prior zooming in.

Response: We have amended the figure and highlighted the mutated residue prior zooming in.

Comment: All dates need to be revised thoroughly, either those of this study at their different occurrences in the manuscript and of other cited work.

Response: We thank the reviewer for the detailed assessment of our dates in this study. To this end, we have edited several sections to better deliver our data, particularly in the methods section:

“The corresponding SARS-CoV-2 sequences of the infected patients were extracted from GISAID and the retrospective clinical histories of the patients matching to the SARS-CoV-2 sequences were collected.” We have also corrected other inconsistent dates within this manuscript.

Reviewer 2

Comment: The manuscript is well written in terms of the language and clarity of statements. However, some words need to be revised and modified for further clarity, notably:

a. Line 34 -> “persistent”.

b. Line 151 -> “Genomic distribution of amino acid substitutions ...”.

c. Line 168 -> “contains”.

d. Line 257 -> “occur”.

e. Line 353 -> “demography”.

Response: We thank the reviewer for these comments and have refined the specified terms to enhance the clarity of our sentences as the following: 

a. Line 34 -> “persistent”.

Revised to: chronic diseases

b. Line 151 -> “Genomic distribution of amino acid substitutions ...”.

Revised to “frequencies of amino acid substitutions”

c. Line 168 -> “contains”.

Revised to “encodes”

d. Line 257 -> “occur”.

Revised by omitting “occur”. The new sentence is now as following: ‘It is possible that comorbidities contribute to the enhanced systemic inflammation releases reactive oxygen species and drives mutations. Changes in the the biochemical process have been shown to induce errors in replication, editing, or damage to a nucleic acid (24).’

e. Line 353 -> “demography”.

Revised by omitting “demography”

Comment: In line 103, There are unresolved abbreviations that need to be elucidated. The abstract should have the full wording of any abbreviations as it is mostly read separately.

Response: We have added full terms for all abbreviations where necessary. Similarly for the abstract, we have included full wordings for the provided abbreviations.

Comment: In line 139 in the Results section, the percentages summarized in text from Table 1 were different than those in the table, so please revise.

Response: We have corrected the numbers in the table to match the text.

Comment: In Figure 1 and more than once in the text, Age and sex were included as important factors in the rate of viral mutation (as in line 114 in the Results and line 259 in the Discussion). However, the significance of these factors was not addressed in the study which was carried out using a dataset with a 2:1 female to male ratio. The authors need to show how sex could impact their results

Response: We thank the reviewer for this insightful comment. To address this, we have include the following input in our discussion:

“Regarding the sex ratio of our dataset, we utilised a convenience sampling approach, thus demonstrating an unbiased scenario of COVID-19 patients of HCTM within the specified time. However, our dataset featured a 2:1 female-to-male ratio. We performed a correlation analysis between gender and the frequency of amino acid substitutions. However, there was no statistically significant association between gender and the number of mutations, although males reported higher mutations than females [p=0.769]. Based on this trend, it is likely that males are predisposed towards generating higher frequencies of SARS-CoV-2 mutations. However, a bigger sample size is warranted to fully comprehend the significance of sex with mutation frequencies for SARS-CoV-2.”

Comment: In line 265 in the Discussion section, the statement “work of ...... supported our findings”, is somehow confusing as the reader here may need to know which findings in particular are consistent with the cited reference.

Response: We apologise for the misleading sentence in this section and have reword the line to the following:

“Work by Maurya et al. (2022) supported our findings, as they identified a single mutation (S194L) to frequently occur in their mortality group, implying the exclusivity or tendency for mutations to occur in patients with severe disease progressions.”

---

## [Decision Letter · Decision Letter 1]

12 Feb 2024

PONE-D-23-28307R1Unravelling the Link between SARS-CoV-2 Mutation Frequencies, Patient Comorbidities, and Structural DynamicsPLOS ONE

Dear Dr. Azmi,

Thank you for submitting your manuscript to PLOS ONE. After careful consideration, we feel that it has merit but does not fully meet PLOS ONE’s publication criteria as it currently stands. Therefore, we invite you to submit a revised version of the manuscript that addresses the points raised during the review process.

Please submit your revised manuscript by Mar 28 2024 11:59PM. If you will need more time than this to complete your revisions, please reply to this message or contact the journal office at plosone@plos.org. Please include the following items when submitting your revised manuscript:A rebuttal letter that responds to each point raised by the academic editor and reviewer(s). You should upload this letter as a separate file labeled 'Response to Reviewers'.A marked-up copy of your manuscript that highlights changes made to the original version. You should upload this as a separate file labeled 'Revised Manuscript with Track Changes'.An unmarked version of your revised paper without tracked changes. You should upload this as a separate file labeled 'Manuscript'.If applicable, we recommend that you deposit your laboratory protocols in protocols.io to enhance the reproducibility of your results. Protocols.io assigns your protocol its own identifier (DOI) so that it can be cited independently in the future. For instructions see: https://journals.plos.org/plosone/s/submission-guidelines#loc-laboratory-protocols. Additionally, PLOS ONE offers an option for publishing peer-reviewed Lab Protocol articles, which describe protocols hosted on protocols.io. Read more information on sharing protocols at https://plos.org/protocols?utm_medium=editorial-email&utm_source=authorletters&utm_campaign=protocols.

We look forward to receiving your revised manuscript.

Kind regards,

Ahmed A. Al-Karmalawy, Ph.D.

Academic Editor

PLOS ONE

Journal Requirements:

Reviewers' comments:

Reviewer's Responses to Questions

**Comments to the Author**

1. If the authors have adequately addressed your comments raised in a previous round of review and you feel that this manuscript is now acceptable for publication, you may indicate that here to bypass the “Comments to the Author” section, enter your conflict of interest statement in the “Confidential to Editor” section, and submit your "Accept" recommendation.

Reviewer #1: (No Response)

Reviewer #2: (No Response)

2. Is the manuscript technically sound, and do the data support the conclusions?

Reviewer #1: (No Response)

Reviewer #2: Yes

3. Has the statistical analysis been performed appropriately and rigorously? 

Reviewer #1: (No Response)

Reviewer #2: Yes

4. Have the authors made all data underlying the findings in their manuscript fully available?

Reviewer #1: (No Response)

Reviewer #2: Yes

5. Is the manuscript presented in an intelligible fashion and written in standard English?

Reviewer #1: (No Response)

Reviewer #2: Yes

6. Review Comments to the Author

Reviewer #1: (No Response)

Reviewer #2: 1. In line 251, WT and the mutant were mixed up. Which one has the lower average RMSD value?

2. In line 371, I highlighted this in the last round, but no response received. I will be clearer. Does the phrase mean that aside from ORF1a P1640L, specifically P1640S, and P1640L again? Isn't the former P1460L the exact latter P1460L?

3. A thorough check of spelling and punctuations are still needed for a best shape.

I again uploaded a highlighted copy of the manuscript for pointing at my comments.

7. PLOS authors have the option to publish the peer review history of their article (what does this mean?). If published, this will include your full peer review and any attached files.

Reviewer #1: **Yes**

Reviewer #2: **Yes**

---

## [Author Response · Author response to Decision Letter 1]

15 Feb 2024

We would like to thank the reviewers in evaluating our manuscript titled “Unravelling the Link between SARS-CoV-2 Mutation Frequencies, Patient Comorbidities, and Structural Dynamics," submitted under Manuscript ID PONE-D-23-28307 to PLOS ONE. Your insightful comments has been helpful in refining our manuscript.

In this rebuttal letter, I would like to highlight the key points raised by the reviewers and outline the specific changes made to address each comment

Comment 1: In line 251, WT and the mutant were mixed up. Which one has the lower average RMSD value?

Response: We thank the reviewer for the comment, and acknowledge that this was an error form our part.

We have fixed this information to include the correct RMSD to the following sentence: 

‘The average RMSD values throughout the entire simulation times for P822L (1.68 nm) was lower compared to WT (2.86 nm). Additionally, since the WT possesses a higher maximum RMSD, the energy landscape and conformational changes within the WT is more significant compared with P822L.’

Comment 2: In line 371, I highlighted this in the last round, but no response received. I will be clearer. Does the phrase mean that aside from ORF1a P1640L, specifically P1640S, and P1640L again? Isn't the former P1460L the exact latter P1460L?

Response: We thank the reviewer for the comment and apologise for missing this note during the first correction. This was again, an oversight from our end and only meant to mention P1640S.

We also fixed the sentence structure to the following: ‘Throughout the virus's evolution, various mutations in the same residue occurred, one of them being ORF1a P1640S (37).’ 

Comment 3: A thorough check of spelling and punctuations are still needed for a best shape. I again uploaded a highlighted copy of the manuscript for pointing at my comments.

Response: We have sought assistance from an English-proof reader to correct the grammatical and sentence structure errors within this manuscript.

We look forward to hearing from you in due time regarding our submission and to respond to any further questions and comments you may have.

---

## [Decision Letter · Decision Letter 2]

26 Feb 2024

Unravelling the Link between SARS-CoV-2 Mutation Frequencies, Patient Comorbidities, and Structural Dynamics

PONE-D-23-28307R2

Dear Dr. Azmi,

We’re pleased to inform you that your manuscript has been judged scientifically suitable for publication and will be formally accepted for publication once it meets all outstanding technical requirements.

Kind regards,

Ahmed A. Al-Karmalawy, Ph.D.

Academic Editor

PLOS ONE

Reviewers' comments:

Reviewer's Responses to Questions

**Comments to the Author**

1. If the authors have adequately addressed your comments raised in a previous round of review and you feel that this manuscript is now acceptable for publication, you may indicate that here to bypass the “Comments to the Author” section, enter your conflict of interest statement in the “Confidential to Editor” section, and submit your "Accept" recommendation.

Reviewer #2: All comments have been addressed

2. Is the manuscript technically sound, and do the data support the conclusions?

Reviewer #2: Yes

3. Has the statistical analysis been performed appropriately and rigorously? 

Reviewer #2: Yes

4. Have the authors made all data underlying the findings in their manuscript fully available?

Reviewer #2: Yes

5. Is the manuscript presented in an intelligible fashion and written in standard English?

Reviewer #2: Yes

6. Review Comments to the Author

Reviewer #2: (No Response)

7. PLOS authors have the option to publish the peer review history of their article (what does this mean?). If published, this will include your full peer review and any attached files.

Reviewer #2: **Yes**

---

## [Editor Report · Acceptance letter]

6 Mar 2024

PONE-D-23-28307R2 

PLOS ONE

Dear Dr. Azmi, 

I'm pleased to inform you that your manuscript has been deemed suitable for publication in PLOS ONE. Congratulations! Your manuscript is now being handed over to our production team.

Kind regards, 

on behalf of

Dr. Ahmed A. Al-Karmalawy 

Academic Editor

PLOS ONE